# Effect of GAN Augmented Dataset Size on Deep Learning-based Ultrasound Bone Segmentation Model Training

## Abstract

Recently, ultrasound imaging is increasingly being used as intra-operative imaging modality in osteolytic bone surgery due to its cost-effectiveness and radiation-free nature. Deep learning approaches have shown remarkable success in segmenting bone surface from ultrasound images. However, limited training dataset size has always hindered the success of deep learning approaches, which is especially evident in ultrasound bone segmentation since there is no publicly available quality dataset. To resolve the issue, in addition to standard data augmentation approaches, generative adversarial networks (GANs) have recently been used for generating augmented training samples. Although the addition of the generative approach in data augmentation is observed to have a positive effect on deep learning architecture's performance, the question about the effect of using a multi-fold of GAN generated training dataset remains unanswered. In this work, we have generated 14 fold of GAN-augmented training dataset and evaluated the performance of the network for successively increased dataset size. Our test results show that although the use of the GAN-augmented training dataset in addition to standard augmentation approaches helps perform the network better, the addition of multi-fold GAN-augmented datasets has no noticeable performance gain.

**Keywords:** Ultrasound bone segmentation, GAN, data augmentation

## 1. Introduction

Ultrasound imaging due to its cost-effectiveness, ease of use, and radiation-free nature is gaining popularity as an intra-operative imaging modality in osteolytic bone surgery where the bone model reconstructed from segmented ultrasound images is registered to pre-operative MRI bone model to establish intra-operative navigation. Although image quality in ultrasound imaging suffers a lot due to speckle noise, shadow effect, and low contrast, recently, deep learning approaches have shown remarkable success in ultrasound bone segmentation (Salehi et al., 2017). However, deep learning approaches require a lot of quality training samples to be effective in segmentation, which is not always possible to acquire in case of medical imaging applications such as ultrasound bone segmentation. This is because the process of acquiring training samples requires domain experts and often is a very tedious job. Therefore, data augmentation is very essential in training of deep learning models for medical imaging applications. However, although very successful, standard augmentation approaches lack the capability to model patient-specific non-dominant features in the augmented samples. Generative adversarial networks (GANs) (Goodfellow et al., 2014) on the other hand has shown the ability to find inherent data distribution to model

these minute changes. This renders GANs a very attractive addition to data augmentation techniques for medical imaging-based deep learning applications.

Although it has been proven that a larger training dataset always has a positive effect on the training of deep learning models (Sun et al., 2017), whether the use of larger training dataset augmented by GANs has the similar effect or not is still an open question. To answer the question, using a multiple-snapshot approach, we have generated a 15 fold of GAN-augmented training dataset and observed the accuracy of the network when the network is trained with an increasing number of training samples for ultrasound bone segmentation model.

## 2. Method

For GAN-based data augmentation for bone segmentation from ultrasound images, image-to-image translations architectures (pix2pix) (Isola et al., 2017) are perfect since the networks could be trained to generate synthetic ultrasound images from the labeled images. However, pix2pix architectures are known to produce very deterministic output from an input. In other words, pix2pix can only generate one set of GAN-augmented dataset from one set of input images. In the multiple-snapshot image-to-image translation approach, the training procedure ensures that there are multiple snapshots available from the training process which are handpicked based on their ability to generate realistic samples. The process is shown in Fig. 1. This approach allowed us to generate a 15-fold GAN-augmented training dataset. Each of the snapshots was handpicked based on its ability to generate realistic ultrasound images from the supplied label image.

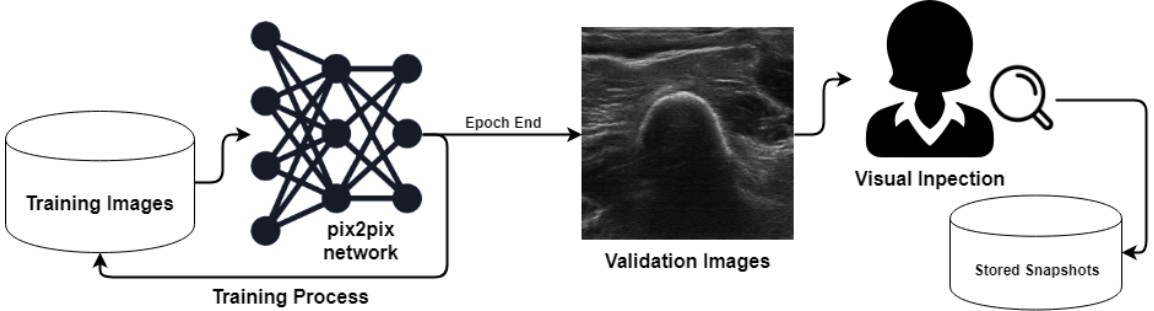

Figure 1: The process of multiple snapshot pix2pix approach

## 3. Experiment & Results

We have used a standard U-Net architecture for performing bone segmentation from ultrasound images. For this experiment, we kept all of the hyperparameters fixed while only changing the dataset size. Standard augmentation approaches such as contrast and brightness enhancement [0.5 1.5], random horizontal flip, rotation [0° 15°], Gaussian blur were applied randomly to each of the training samples. These standard augmentation approaches were selected from observing the acquisition process of ultrasound images for bone imaging.

The network was trained on 3,104 ultrasound images from the humerus bone region from 17 volunteers while tested on 870 ultrasound images from the humerus bone region from 5 volunteers as well as 1,200 ultrasound images from tibia and femur bone region from the same 5 volunteers. We represent the humerus bone region as 'trained region' and tibia and femur bone region as 'untrained region'. The result is shown in Fig. 2.

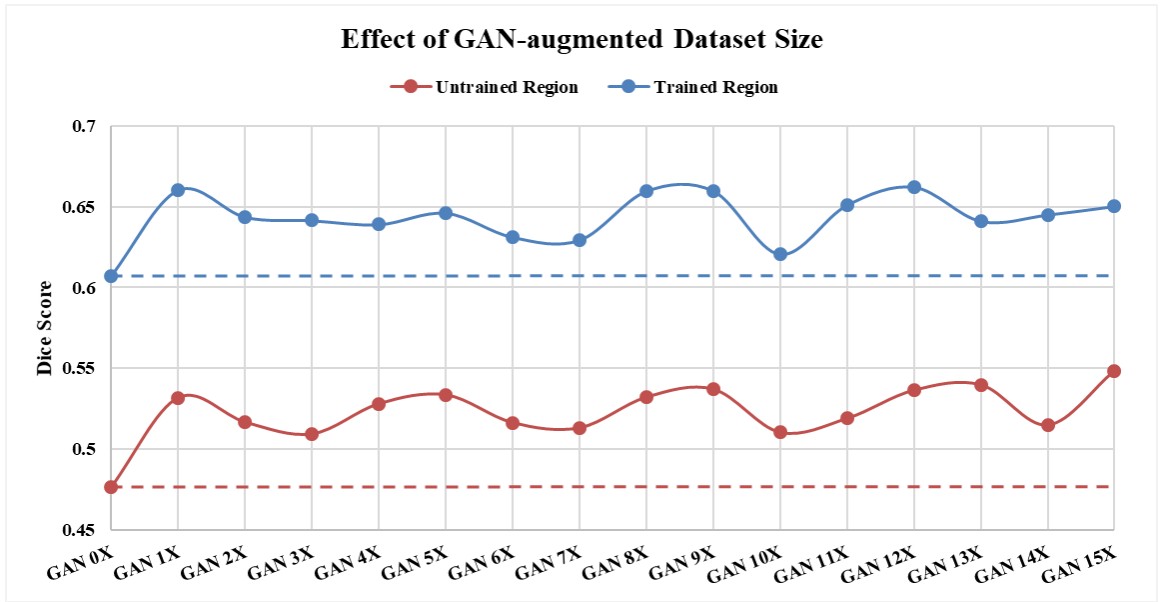

Figure 2: Effect of GAN-augmented dataset size in ultrasound segmentation on both trained and untrained regions. Here, 'GAN0X' means no GAN-augmented images are used while other represents the number of GAN-augmented dataset used for training process

## 4. Conclusion

The result shown in Fig. 2 reveals that, although the GAN-augmented dataset has a positive effect on the network's accuracy, adding more GAN-augmented images generated from the same dataset does not contribute to better accuracy. This trend is prominent in both trained and untrained region. In our visual inspection, a multi-fold GAN-augmented dataset generated from multiple snapshot pix2pix approach offered patient-specific non-dominant feature variations, although the results suggest that the variations are not enough to account for better accuracy. However, the demonstration only includes a specific application area with a specific algorithm to generate a multi-fold GAN-augmented dataset, which suggests that there is more to explore to answer the question.

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
