# OpenReview forum: "Effect of GAN augmented dataset size on deep learning-based ultrasound bone segmentation model training"
_MIDL.io/2020/Conference — Submitted to MIDL 2020_

### Official Review · AnonReviewer2 · 2020-02-24
**Important topic but weak evaluation**

**Rating:** 2
**Confidence:** 5

**Review:**

Segmentation of bone surfaces from intra-procedure ultrasound data is an important step in ultrasound-guided surgical and non-surgical procedures. However, this is a challenging task as low signal to noise ratio, imaging artifacts, user dependent imaging result in large segmentation errors for traditional image analysis methods developed to provide a solution to this challenging task. Most recently deep learning-based approaches have been investigated by various researchers. However, as the authors mention, scarcity of bone ultrasound data for training hinders the widespread adaptability of these methods. The manuscript tries to investigate the effect of GAN for simulation of ultrasound data from manual segmentation to improve the training data size. Although the work tries to address, and important problem important details are missing. Major and minor points are provided below.


Major

The authors do not provide any example images on what kind of bone ultrasound data was used for training or generating. High quality ultrasound data corresponds to high intensity bone interfaces followed by low intensity bone shadow artifact. If these two important features are present in the data collected for testing I don’t think a well-trained (using the ~3,000 scans for example) Unet would have low error values. Usually Unet fails if the collected ultrasound data is of low quality: Low intensity blurred bone boundaries, artifacts in the bone shadow region). This is usually the result of wrong orientation of the ultrasound transducer with respect to the imaged bone anatomy or when imaging complex shape bone surfaces such as the spine. The work would be of stronger impact if GAN are used to generate low quality ultrasound data. In summary more details about what king of data was used is missing.

I was having a hard time to understand what the terminology used for GAN1X GAN 2X…. GAN 15X stand for? Does GAN2X mean the pix2pix GAN architecture generates twice the size of the trained data? This should be clearly explained.

Only single evaluation metric was investigated. Sensitivity, specificity, F-score, average Euclidean distance (usually used to report the localization accuracy of bone segmentation) was not investigated.

Reported DICE values are very low. The authors are not discussing this. Was there a specific reason for this? The data size used for training is not too small! As a follow up comment: Most recent state of the art methods for bone segmentation, based on deep learning, have achieved significantly improved results over Unet (see below for references). These methods combine multi-feature images as an input to Unet together with the traditional B-mode ultrasound data. The authors should include these and discuss how if the reported dice values could be improved using multi-feature CNN architecture.

Wang P, Patel VM, Hacihaliloglu I. Simultaneous segmentation and classification of bone surfaces from ultrasound using a multi-feature guided CNN. InInternational Conference on Medical Image Computing and Computer-Assisted Intervention 2018 Sep 16 (pp. 134-142). Springer, Cham.

Alsinan AZ, Patel VM, Hacihaliloglu I. Automatic segmentation of bone surfaces from ultrasound using a filter-layer-guided CNN. International journal of computer assisted radiology and surgery. 2019 May 1;14(5):775-83.

El-Hariri H, Mulpuri K, Hodgson A, Garbi R. Comparative Evaluation of Hand-Engineered and Deep-Learned Features for Neonatal Hip Bone Segmentation in Ultrasound. InInternational Conference on Medical Image Computing and Computer-Assisted Intervention 2019 Oct 13 (pp. 12-20). Springer, Cham.

---

### Official Review · AnonReviewer3 · 2020-03-13

**Rating:** 2
**Confidence:** 5

**Review:**

The authors evaluate the effect of data augmentation -- generated using a 15 fold pix-to-pix network -- in improving an Ultrasound Bone Segmentation Model.
They show while having data augmentation helps perse, the multifold aspect of it is not useful.
There are some major concerns with the paper which I will list as follows:
1) the pix-to-pix network is not a generative model but rather an image translation model. There it's a one to one mapping network. There are variations of that model that augment the pix2pix with a stochastic variable which one can sample from. E.G Toward Multimodal Image-to-Image Translation.  I recommend the authors to include this model in the evaluation since its a simple modification to the pix2pix network.

2) While the authors do evaluate the effect that data generation has on the downstream task (segmentation task) they do not evaluate the image translation models. This could be done qualitatively by showing, conditioned on a segmentation mask,  how the images from different image translation networks look like and what degrees of variation we can hope to achieve using the 15 fold framework.

Based on these comments I don't see the paper ready to be presented as is. I encourage the authors to improve the submission and try again.

---

### Official Review · AnonReviewer4 · 2020-03-14

**Rating:** 2
**Confidence:** 3

**Review:**

pros: This paper tries to discuss if the GAN-based data augmentation could improve the bone segmentation from ultrasound images.

cons: 1) The paper fails to give detail about why and how the visual inspection is introduced. According to the authors, the inspectors only make sure the ultrasound looks real, then store them as snapshots. How do these snapshots improve the segmentation performance? Are the generated images have been checked with the ground truth label images?
2) GAN is dynamically updated and balanced during the training. I do not see a strong motivation to add human interaction to augment the dataset.

---

### Meta-Review · Area_Chair1 · 2020-03-31
**MetaReview of Paper227 by AreaChair1**

**Rating:** 2

**Metareview:**

This paper investigate if the use of GAN-based data augmentation improves bone segmentation from an ultrasound image. The three reviewers consistently rate the paper as 'weak reject' due to reasons like weak evaluation, lack of motivation, etc. The AC agrees with the three reviewers and encourages the authors to follow the suggestions from the reviewers to improve the paper quality.

**Paper Type:**

methodological development

---

### Decision · Program_Chairs · 2020-04-11

Reject